# Role of Ketamine in Multimodal Analgesia Protocol for Bariatric Surgery

**DOI:** 10.3390/medicina56030096

**Published:** 2020-02-26

**Authors:** Greta Kasputytė, Aurika Karbonskienė, Andrius Macas, Almantas Maleckas

**Affiliations:** 1Department of Anaesthesiology, Lithuanian University of Health Sciences, 50161 Kaunas, Lithuania; aurika.karbonskiene@kaunoklinikos.lt (A.K.); andrius.macas@kaunoklinikos.lt (A.M.); 2Department of Surgery, Lithuanian University of Health Sciences, 50161 Kaunas, Lithuania; almantas.maleckas@yahoo.com

**Keywords:** postoperative pain, ketamine, remifentanil, bariatric surgery

## Abstract

*Background and Objectives:* Acute postoperative pain is one of the most undesirable experiences for a patient in the postoperative period. Many options are available for the treatment of postoperative pain. One of the methods of multimodal analgesia is a combination of opioids and adjuvant agents, such as ketamine. The aim of this study was to evaluate the effect of a pre-incisional single injection of low-dose ketamine on postoperative pain after remifentanil infusion in patients undergoing laparoscopic gastric bypass or gastric plication surgery. *Materials and Methods:* The prospective, randomized, double-blinded and placebo-controlled trial took place at the Hospital of the Lithuanian University of Health sciences KaunoKlinikos in 2015–2017. A total of 32 bariatric patients (9 men and 23 women) were randomly assigned to receive a single pre-incisional injection of ketamine (0.15 mg/kg (LBM)) (ketamine, K group) or saline (placebo, S group). Standardized protocol of anesthesia and postoperative pain management was followed for all patients. Postoperative pain intensity, postoperative morphine requirements, incidence of side effects and patients’ satisfaction with postoperative analgesia were recorded. *Results:* Thirty-two patients undergoing bariatric surgery: 18 (56.25%; gastric bypass) and 14 (43.75%; gastric plication) were examined. Both groups did not differ in demographic values, duration of surgery and anesthesia and intraoperative drug consumption. Postoperative pain scores were similar in both groups (*p* = 0.105–0.941). Morphine consumption was 10.0 (7.0–12.5 mg) in group S and 9.0 (3.0–15.0 mg) in group K (*p* = 0.022). The incidence of side effects was similar in both groups (*p* = 0.412). Both groups demonstrated very high satisfaction with postoperative analgesia. *Conclusions:* Pre-incisional single dose ketamine reduces postoperative opioids consumption, but does not have an effect of postoperative pain intensity and side effects after remifentanil infusions. Very high patient satisfaction is achieved if standard multimodal analgesia protocol with an individual assessment of pain and dosage of medications is followed.

## 1. Introduction

Acute postoperative pain is one of the most undesirable experiences for patients in the postoperative period [1]. Effective pain management after surgery provides conditions for full early rehabilitation. Patients are more active, thus expecting a lower occurrence of thrombotic, respiratory and other complications after surgery [2]. Good pain management is especially important in patients at higher risk for these complications, such as those with morbid obesity. Many options are available for the treatment of postoperative pain [3]. One of the methods of multimodal analgesia is a combination of opioids and adjuvant agents, such as ketamine [4]. 

## 2. Materials and Methods

The prospective, randomized, double-blinded and placebo-controlled trial took place at the Hospital of the Lithuanian University of Health Science KaunoKlinikos in 2015–2017. After approval of the local Ethics Committee (2015-04-08, BEC–MF–713) and with the patients’ written consent, 32 morbidly obese patients were enrolled in the study. The inclusion criteria were: II or III ASA physical status, age >18 years, bariatric surgery with general remifentanil anesthesia. Patients with anamnesis of using opioids for the treatment of chronic pain and opioid dependence, younger than 18 years, were excluded from the study.

The patients were randomly assigned to receive a single pre-incisional injection of ketamine (0.15 mg/kg) (lean body mass, LBM) (ketamine, K group) or saline (placebo, S group). Dosage of medications was based on the patient lean body mass.

In the operating room standard monitors were used: ECG, non-invasive blood pressure, pulse oximetry and EtCO_2_.

Anesthesia was induced after pre-oxygenation by face mask with 100% oxygen with intravenous (i.v.) fentanyl 1.2–1.5 mcg/kg, i.v. propofol 2 mg/kg and i.v. cisatracurium 0.1 mg/kg.

Anesthesia was maintained with sevoflurane 0.7–0.8 MAC, remifentanil 0.2–0.5 mcg/kg/min and noradrenaline 0.01–0.04 mcg/kg/min, based on heart rate and blood pressure stability according to judgement of an experienced anesthesiologist (to individual patients’ response). 

The lungs were ventilated with a mixture of 60% of oxygen and air in the pressure control mode, to maintain normocapnia (EtCO_2_ 35–45 mmHg).

All patients received a total volume of infusion of 500 mL Ringer’s Lactate and 500 mL Dextran solutions during anesthesia.

In the operating room the postoperative analgesia was performed by infiltration of 0.25% bupivacaine into laparoscopic trocars’ insertion site, i.v. acetominofenum 1000 mg and i.v. ketoprofenum 100 mg before the end of surgery and i.v. morphine 5 mg after the end of remifentanil and noradrenaline infusions. 

All patients were extubated in the operating room and sent to postoperative care unit (PACU).

Standard post-operative monitoring was applied: ECG, non-invasive blood pressure, pulse oximetry. All patients were administered 2–4 l/min oxygen by nasal cannula. Patients were monitored in PACU for 2.5 h and pain intensity was recorded every 15 min. Postoperative pain was treated with boluses of i.v. morphine (3 mg) at 3 min intervals on request, if pain intensity exceeded the score of 5 according to the numeric pain rating scale (NPRS). After the transfer of patients’ to regular unit, pain intensity was recorded every 6 h. 

General pain characteristics, duration of surgery and anesthesia, pain intensity, postoperative morphine requirements, incidence of side effects and patient satisfaction with postoperative analgesia were recorded. 

Statistical analyses were performed using Statistical Package for Social Sciences (SPSS) software version 23.0. Continuous variables were presented as median ± min/max values. Categorical variables are summarized as frequencies. Differences in continuous variables between the two groups were evaluated using the Mann–Whitney U test. Categorical variables were compared using the Pearson χ^2^ test. A *p*-value less than 0.05 was considered significant. 

## 3. Results

Thirty-two patients were enrolled in the study (Figure 1).

Patient demographic data are shown in Table 1. There were no differences in age, sex, body mass index, lean body mass, duration of anesthesia, surgery and pneumoperitoneum, remifentanil and noradrenaline consumptions between the two groups. 

The total amount of morphine during the first 2.5 h after surgery is shown in Table 2.

During the study, we assessed the relationship between the pain score and total morphine dose. We found strong relation in the ketamine group, *r* (Spearman) = 0.818, *p* < 0.001 and no relation in the saline group, *r* (Spearman) = 0.19, *p* = 0.465 (Figure 2).

In assessing the postoperative pain of the subjects, the time of arrival in the post-anesthetic care unit was chosen as the first point. Later, pain was evaluated at intervals of 15 min for the first 135 min. When comparing the NPRS scores of the patients in the control and research groups, we did not find a statistically significant difference at either measurement point (*p* = 0.105–0.941) (Figure 3). 

While in the PACU, patients were monitored for side effects. No urinary retention, itching, respiratory depression, agitation or hallucinations were observed in patients. The incidence of other undesirable effects after surgery was not different between the two groups (*p* = 0.412) (Figure 4).

We did not notice ketamine side effects (agitation, hallucinations) in any of the patients. 

Patients evaluated the satisfaction of pain management the morning after surgery. The subject could express their level of satisfaction in five possible levels, from “very satisfied” to “very dissatisfied”. Patient satisfaction with pain treatment was not different between the two groups (*p* = 0.21) (Figure 5). 

We also analyzed whether patients would choose the same type of pain treatment if they need to repeat the surgery or would they recommend it to their loved ones. No statistically significant difference was found between the two groups (*p* = 0.47 and *p* = 0.47, respectively).

During the research, we found a positive relationship between:Total morphine consumption and patient satisfaction. The higher the morphine dose was administered for patients in PACU, the higher their satisfaction with postoperative pain management was, *r* (Spearman) = 0.36 (*p* = 0.045).Postoperative pain intensity and patient satisfaction. The more severe the pain was experienced by the patients, the more satisfied they were with analgesia, *r* (Spearman) = 0.51 (*p* = 0.003).

## 4. Discussion

Sufficient pain management is critical for full early rehabilitation and associated better outcomes after surgery. During the study, we found that in the treatment of pain after laparoscopic gastric bypass and gastric plication surgeries, the required morphine dose in both groups was low (from 3 to 15 mg). Nevertheless, in our study, pain was managed effectively (the NPRS median during the observation time ranged from zero to four). We assume that this was affected by active patient monitoring (every 15 min), individual attention and communication with the subjects in the PACU. A similar total morphine amount used in the treatment of postoperative pain for the first two hours, in the PACU, after bariatric surgeries was demonstrated by Hasanein et al. in the study in 2011 [5]. Morphine consumption (mg) ranged from 10.4 ± 1.4 in the control group to 6.3 ± 10.6 in the study group (*p* < 0.05). Saumier and other authors in the study published in 2013 [6] also presented the results of total morphine consumption used in the PACU after laparoscopic bariatric surgery. In France, as in Lithuania, the amount of morphine used in the PACU was not high, ranging from 1 to 19 mg. In some populations, the morphine needed after surgery is higher: in a study conducted at the Akram Medical Centre Hospital in Tehran, Iran, the mean morphine dose (mg) after gastric bypass surgery was 32.5 ± 14.1 in the control group and 20.7 ± 13.7 in the study group (*p* = 0.08) [7]. 

We found that the need for morphine in postoperative patients who received ketamine was lower than in patients who have not received it (*p* = 0.022). The same results are presented by other authors in various operations [8,9]. We also found that the total morphine dose used was directly proportional to the intensity of postoperative pain, *r* (Spearman) = 0.396 (*p* = 0.025). This has been found by other authors following traumatology surgery [10]. We wanted to find out whether low-dose ketamine affects the relationship between pain and morphine consumption, therefore, we compared individual groups. This direct connection was revealed only in the research group, *r* (Spearman) = 0.396 (*p* = 0.025). Such results, potentially, indicate that ketamine, acting through the NMDA and opioid receptors, provides favorable conditions for adequate postoperative pain management. 

The undesirable post-anesthetic morphine-related effects and adverse reactions were not common. In the study, we found that 34% of patients were nauseous and only 3% of them vomited. We believe that such favorable results were due to careful observance of postoperative nausea and vomiting (PONV) prophylaxis recommendations. We found only one clinical study where, in order to evaluate the efficiency of PONV prevention medicines and their combinations, the patients in the control groups did not receive any PONV prevention medications at all. It was found that the combination of ondansetron (4 mg) and dexamethasone (8 mg) reduced the incidence of POVN from 85% (placebo group) up to 10% (research group) (*p* = 0.0001) [11]. We did not find undesirable effects related to ketamine. We think that they were avoided because we chose a very low dose of medication of 0.15 mg/kg (LBM).

Patient satisfaction with analgesia was also researched. Most of the patients (84.4%) of our study were highly satisfied with postoperative pain treatment. Only one patient (3.1%) was very dissatisfied with our services on the next day after operation. After analyzing this case, we noticed that even before the operation he had a panic attack. We believe that this affected the inadequate sensation of pain and excessive dissatisfaction with postoperative analgesia. This is confirmed by the systematic analysis published by Vivian et al. [12] in 2009, where anxiety before various types of surgeries in seen as the most common factor determining the postoperative pain intensity.

It is a paradox, but we found a direct relationship between patient satisfaction and postoperative pain intensity. The stronger the pain, the more satisfied patents were with analgesia, *r* (Spearman) = 0.51 (*p* = 0.003). This phenomenon has been described by anesthesiologists at John Hopkins University in Baltimore in 2012. They found that patient satisfaction was higher when medical staff make all their efforts to reduce postoperative pain rather than the methods in which the pain is actively managed [13]. 

Our study involved only 32 patients, and, therefore, the research was limited to a small sample, which was determined by the number of patients treated in our hospital. Starting our study, we opted to enroll all consecutive bariatric patients in our facility. Eight patients were excluded from the study. The reason for exclusion was non-standard surgical technique: these patients have undergone previous abdominal surgical procedures, resulting in difficulties of surgical access and significantly longer duration of bariatric surgery (167.5 ± 63.7 min, in comparison to 90 ± 19.7 min in standard technique patients). We felt that these patients were not eligible for the analysis due to higher intraoperative consumption of anesthetic medications. 

To highlight the role of ketamine, a larger study should be performed using other, higher doses and different administration routes (continuous infusion, multiple doses during/after surgery). We believe this could be the subject of further research in our clinic. 

## 5. Conclusions

Pre-incisional administration of a single 0.15 mg/kg dose of ketamine reduces postoperative opioids consumption, but does not have an effect of postoperative pain intensity and side effects after remifentanil infusions. Very high patient satisfaction is achieved if standard multimodal analgesia protocol with an individual assessment of pain and dosage of medications is followed.

## Figures and Tables

**Figure 1 medicina-56-00096-f001:**
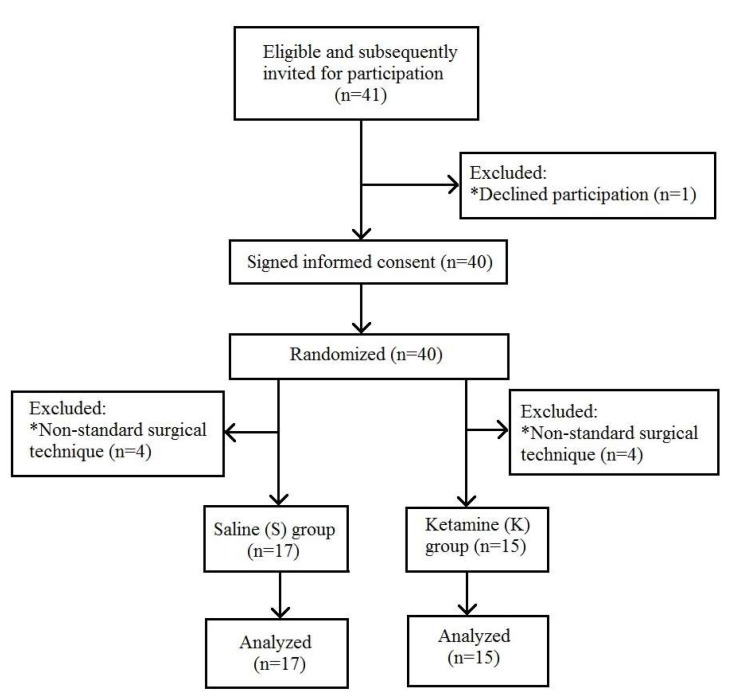
Flow chart for patient enrollment. * Patients who have undergone previous abdominal surgical procedures.

**Figure 2 medicina-56-00096-f002:**
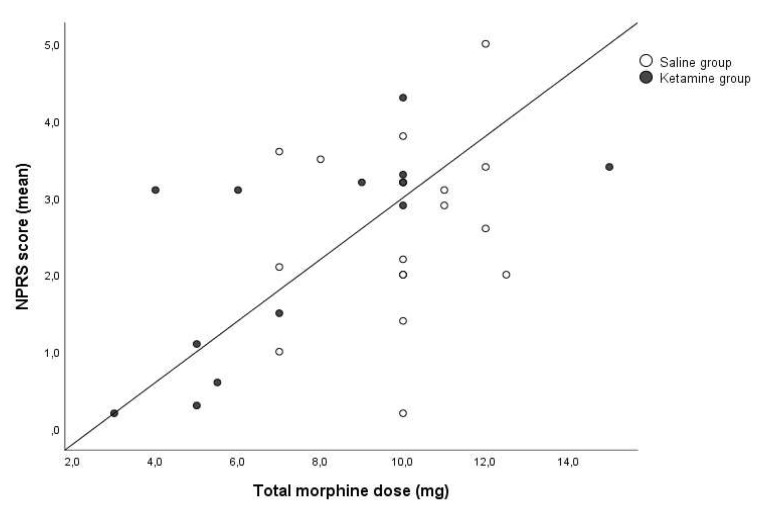
Analysis of the relation of total morphine amount used in the post-anesthetic care unit and patient postoperative pain in the numeric pain rating scale (NPRS) score.

**Figure 3 medicina-56-00096-f003:**
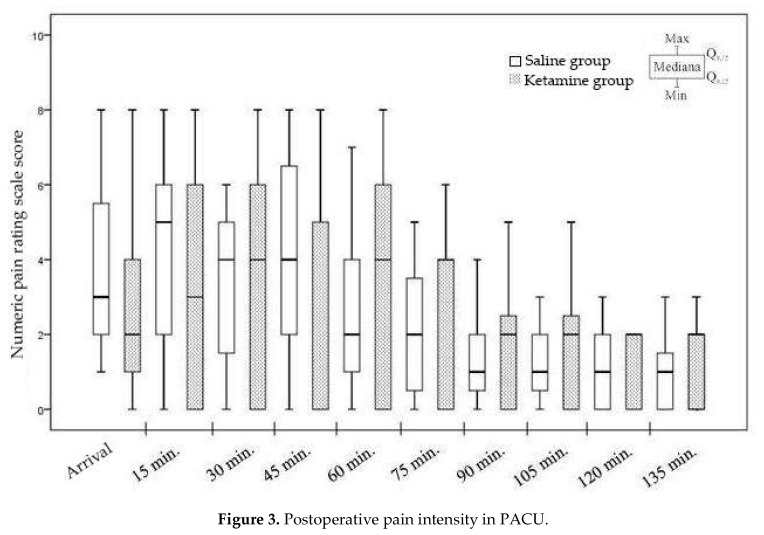
Postoperative pain intensity in PACU.

**Figure 4 medicina-56-00096-f004:**
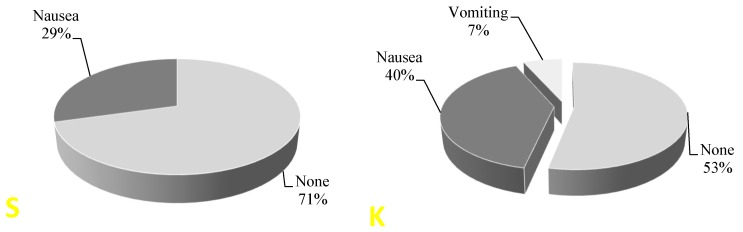
The incidence of side effects after surgery in PACU: S—saline group, K—ketamine group.

**Figure 5 medicina-56-00096-f005:**
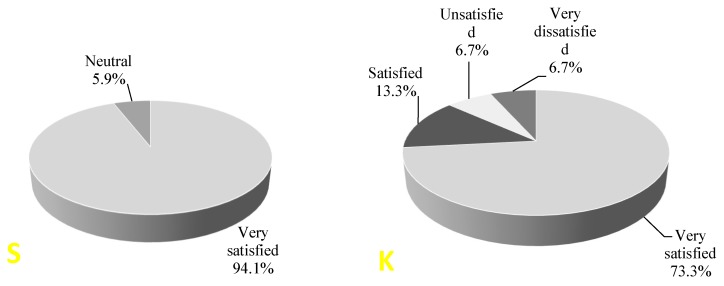
Patient satisfaction with pain treatment in the next day after surgery: S—saline group, K—ketamine group.

**Table 1 medicina-56-00096-t001:** Characteristics of study population.

Variables	Saline (S) Group (*n* = 17)	Ketamine (K) Group (*n* = 15)	*p*-Value
Age (years)			
Median (min–max)	46 (28–64)	45 (18–66)	0.41
BMI (kg/m^2^)			
Median (min–max)	45.0 (34.6–55.4)	43.9 (34.9–55.3)	0.68
LBM (kg)			
Median (min–max)	86.7 (71.2–103.8)	81.7 (74.9–100.5)	0.94
Sex			
Male, *n* (%)	5 (29.4)	4 (26.7)	0.86*
Female, *n* (%)	12 (70.6)	11 (73.3)	
Anesthesia time (min)			
Median (min–max)	112 (95–165)	112 (95–150)	0.53
Surgery time (min)			
Median (min–max)	89 (56.0–144.0)	78.0 (59.0–115.0)	0.33
Pneumoperitoneum time (min)			
Median (min–max)	76 (45.0–114.0)	71.0 (57.0–100.0)	0.50
Consumption of remifentanil (mg)			
Median (min–max)	1.8 (0.75–3.0)	2.0 (0.5–3.2)	0.94
Consumption noradrenaline (mg)			
Median (min–max)	0.4 (0.00–0.8)	0.4 (0.00–1.0)	0.68

*p*-value—Mann–Whitney test, *Pearson χ^2^ test, results are not statistically significant.

**Table 2 medicina-56-00096-t002:** Postoperative morphine consumption.

Variables	Saline (S) Group (*n* = 17)	Ketamine (K) Group (*n* = 15)	*p*-Value
Morphine consumption (mg)			
Median (min–max) Quartiles (Q1–Q3)	10.0 (7.0–12.5) (10.0–11.0)	9.0 (3.0–15.0) (5.25–10.0)	0.022

*p*-value—Mann–Whitney test, results are statistically significant.

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
