# Peer review of "Role of Ketamine in Multimodal Analgesia Protocol for Bariatric Surgery"

_medicina, 2020, doi:10.3390/medicina56030096_

Round 1

Reviewer 1 Report

Good changes. Graphics are clearer. Additional text makes things clearer.  Nice work. 

Reviewer 2 Report

The article can be published in its present form.

Reviewer 3 Report

The text that was added Line 134-140 although could be better placed and incorporated into the discussion. It seems out of place adding it at the beginning of the paragraph

Author Response

This manuscript is a resubmission of an earlier submission. The following is a list of the peer review reports and author responses from that submission.

Round 1

Reviewer 1 Report

The use of ketamine has been shown to reduce the incidence of opioid consumption in many different procedures. This article is similar to other published works for other surgical procedures.

Ketamine at a dose of 0.1-0.5 mg.kg bolus has been shown to reduce opioids consumption, but not an effective means to provide a prolonged analgesic benefit in a number of procedures.  

I appreciate the explanation for the result on Page 6, #2. “The more severe the pain was experienced by the patient’s, the more satisfied they were with analgesia”

For further research would consider a combination of intraoperative ketamine and precedex infusion. vs precedex alone.

Reviewer 2 Report

If this is to be published in an English language journal, it needs some minor grammar and punctuation editing. There appear to be some translation issues with the text. 

The findings are not that novel or surprising.  The authors used an extremely low dose of Ketamine as a single bolus and it was not significant.  There are many similar articles that have shown the same results.  There may be some interest in the use of Ketamine in bariatric surgery patients?  The Ketamine did appear to reduce the amount of post-operative opioids used without a difference in pain satisfaction scores.  This would be great if there was a reduction in the side effects due to opioids but there wasn’t. 

The materials and methods section is choppy and some of the paragraphs could be blended together to make it less so. 

Page 2, lines 56-60. The anesthesia “protocol” is a bit confusing.  Patients received either Ketamine or Saline pre-incision.  All patients received 1.2-1.5 mcg/kg Fentanyl on induction and then anesthesia was maintained with Sevoflurane and Remifentanil.  Then everyone received 1000mg Acetaminophen, 100mg, and IV Morphine 5mg basically upon emergence.  The surgeons also placed local anesthesia in the trocar sites.  That’s a really nice multi-modal analgesic plan.  I’m really not surprised that the very low dose of Ketamine given did not show any significant effects with all the great analgesics already in the anesthetic plan. 

Page 4, lines 96-97.  These three lines of text read strangely. The first sentence should simply state that they assessed the relationship between the pain score and total morphine dose.  The second sentence can say the relationship was strong for the Ketamine group and not so for the control group with the numbers in appropriate spots.  This is a strange finding? 

Figure 2. Add the dots for the control group as solid dots so the reader can see the difference between the ketamine and saline group.  Don’t just tell us.  This is a very odd finding. It’s almost as if the pain assessment tool worked on the Ketamine group and it did not work on the control (saline) group.  That’s an issue for the results. 

Page 5, lines 109-111 tell us what we need to know about side effects. I do not think figure 4 is needed.  If the authors decide to keep figure 4, they need to replace the “A” and “b” letters below with “K” for Ketamine group and “S” for saline group.  Labeling the figires a & b just adds confusion. 

Page 5, line 115. Instead of saying “patients evaluated the satisfaction of pain treatment the next day after surgery, in the morning” consider “Patients evaluated the satisfaction of pain management the morning after surgery”. 

Figure 5. Again, change the letter labels to K and S. 

Page 7, Lines 187-190. I think the initial sentence needs to be modified slightly.  The authors used a REALLY low dose of Ketamine.  0.15 mg/kg is discussed in the literature but most of those studies find either no or a weak benefit.  Using the weights reported in table 1, the average patient in the ketamine group received 12mg (11.25 – 15mg range).  It is very common at our institution to administer 30-50mg as a single preoperative dose to most patients undergoing general anesthesia.  Instead of saying “Pre-incisional single dose ketamine”….. it may be better to quantify the dose somehow.  “Pre-incisional administration of a single “low” dose of ketamine …….  Maybe even listing the dose as “Pre-incisional administration of a single 0.15mg/kg dose of Ketamine”…..  This will allow the reader to better understand the results.  The current statement seems to misrepresent the findings. 

Reviewer 3 Report

The paper presents the results of a prospective randomized double blind trial and thus is of high value. It assesses the usefulness of single dose ketamine at the initiation of general anestesia. 

The methodology and results analysis are sound and valuable.

Some minor remarks are mentioned below:

The abstract should be corrected (p = 0.022) instead of (p = 0.22):

"Postoperative pain scores were similar in both groups (p = 0, 105 26 – 0, 941). Morphine consumption was 10,0 (7,0 – 12,5 mg) in group S and 9,0 (3,0 – 15,0 mg) in group 27 K (p = 0,22)."

Figure 1. includes remarks:  "Excluded: * Non-standard surgical technique (n=4) 

The remark and * require explanation in Figure 1 description and in the Discussion

Minor language errors would benefit from correction, eg.:

"One of the method of multimodal analgesia"